# Evaluation of Anti-Neuroinflammatory Activity of Isatin Derivatives in Activated Microglia

**DOI:** 10.3390/molecules28124882

**Published:** 2023-06-20

**Authors:** Alejandro Cenalmor, Elena Pascual, Sergio Gil-Manso, Rafael Correa-Rocha, José Ramón Suárez, Isabel García-Álvarez

**Affiliations:** 1Facultad de Ciencias Experimentales, Universidad Francisco de Vitoria, Pozuelo de Alarcón, 28223 Madrid, Spain; alex.cenalmor8@gmail.com (A.C.); elena.pascual@ufv.es (E.P.); 2Laboratory of Immune-Regulation, Gregorio Marañón Health Research Institute (IISGM), 28009 Madrid, Spain; sergio.gil@iisgm.com (S.G.-M.); rafael.correa@iisgm.com (R.C.-R.); 3Departamento de Química en Ciencias Farmacéuticas, Facultad de Farmacia, Universidad Complutense de Madrid, 28040 Madrid, Spain; josersua@ucm.es

**Keywords:** neuroinflammation, microglia, isatin, nitric oxide, interleukin 6, TNF-α

## Abstract

Neuroinflammation plays a crucial role in the progression of Alzheimer’s disease and other neurodegenerative disorders. Overactivated microglia cause neurotoxicity and prolong the inflammatory response in many neuropathologies. In this study, we have synthesised a series of isatin derivatives to evaluate their anti-neuroinflammatory potential using lipopolysaccharide activated microglia as a cell model. We explored four different substitutions of the isatin moiety by testing their anti-neuroinflammatory activity on BV2 microglia cells. Based on the low cytotoxicity and the activity in reducing the release of nitric oxide, pro-inflammatory interleukin 6 and tumour necrosis factor α by microglial cells, the *N^1^*-alkylated compound **10** and the chlorinated **20** showed the best results at 25 µM. Taken together, the data suggest that **10** and **20** are promising lead compounds for developing new neuroprotective agents.

## 1. Introduction

In recent years, with the advent of an ageing society, the increase in the number of patients suffering from neurodegenerative disorders, the most common one being Alzheimer’s disease, has been a concern [1]. Although pathological processes underlying neurodegeneration are not fully understood, growing evidence indicates that immune responses, including neuroinflammation, are directly implicated in the processes of neurodegeneration [2]. Among the innate immune cells, microglia are the primary players in neuroinflammation. In neurodegenerative processes, chronically activated microglia release inflammatory cytokines, such as tumour necrosis factor α (TNF-α), interleukin 6 (IL-6) and nitric oxide (NO) [3]. Modulating the microglial response during disease pathology could represent a strategy for developing therapies aimed at counteracting brain degeneration in multiple sclerosis, Alzheimer’s disease, Parkinson’s disease and amyotrophic lateral sclerosis [1,3]. In this sense, developing novel anti-neuroinflammatory agents with a minimum number of synthetic steps is a major challenge to chemists.

Isatin (1*H* indole 2,3-dione) is an endogenous indole present in humans and different plants [4] with significant importance in medicinal chemistry as a precursor for a large number of pharmacologically active compounds [5].

This study focuses on the isatin core and its derivatives, because of their diverse pharmacological activities, including anti-inflammatories, which have not yet been assayed on BV2 microglia cells [6,7]. Moreover, we take advantage of the gentle structure of isatins, such as NH at position 1 and carbonyl functions at positions 2 and 3, to design biologically active analogues capable of controlling microglial activation and the release of proinflammatory mediators. Therapeutic outcomes of isatin and its derivatives against multiple diseases have been recently reviewed and confirm the multitarget profile of isatin analogues and thus, their importance in the field of medicinal chemistry as potent chemotherapeutic agents [8]. However, their anti-inflammatory potential has been described in a few works aimed to evaluate general inflammation, and their anti-neuroinflammatory activities have not been exploited.

The present work describes the synthesis and evaluation of the anti-neuroinflammatory activity of a series of isatin derivatives using lipopolysaccharide (LPS)-activated microglia as a cellular model of murine neuroinflammation. LPS activates BV2 microglia cells, re-creating neuroinflammatory conditions by the activation of the nuclear factor-kappa B (NF-κB) and mitogen-activated protein kinases (MAPKs). The release of proinflammatory cytokines, including IL-6 and TNF-α, as well as the production of NO, is increased. This is a suitable cell model of murine microglia inflammation since 90% of the genes induced by LPS in BV2 microglia cells are also induced in primary microglia and it has been applied previously as murine neuroinflammation cell model as an alternative to animal experimentation [9,10,11,12]. 

## 2. Results and Discussion

### 2.1. Biological Activity of Halogenated Isatin Derivatives

We first evaluated the effect of a series of halogenated isatin derivatives over inflammation in activated microglia and compared it with isatin as a reference compound. Based on the reported analysis of the structure–activity relationship (SAR) for the potential anti-inflammatory activity of isatins [8], we selected analogues bearing small substituents at the *N*-position and at the aryl ring to perform initial anti-neuroinflammatory screening (Table 1). 

The substitution at the 5-position of isatin (compound **1**) with fluor, chloro and bromo (compounds **2**, **3** and **4**, respectively) increases the liphophilicity of compounds as reflected by their Log *p* values. Moreover, the alkylation of the nitrogen with a methyl group (compounds **5** and **6**) lead to higher liphophilicity in comparison with their analogues without the methyl group at position 1 and limits the formation of hydrogen bonds in this position, which might affect the biological activity. Compounds **7** and **8** present substitutions at the 4-position of isatin with chloro and bromo, respectively. Finally, we selected compound **9** as an example of disubstitution at the aryl ring.

We measure NO release by LPS-activated BV2 microglia cells, and to discard the possibility that the inhibition of NO release was affected by the cytotoxicity of compounds, cell viability was assessed by MTT. Cells were preincubated with isatin derivatives for 1 h before the LPS stimulation. We used 1 µg/mL of LPS to activate BV2 cells, since this concentration can stimulate inflammatory conditions [13] and we observed the same cell viability as control cells (Table 1). NO is a key mediator of neurotoxicity with multiple roles in the central nervous system. NO is produced by a variety of cell types, including macrophages and microglia, in response to inflammatory stimuli [14]. The release of NO was determined by measuring concentration of nitrite, a marker of NO production formed by its oxidation, in cell supernatants. We used sulfanilamide and naphthylethylenediamine to measure nitrite concentration, which is currently the most frequently used Griess assay in nitrite quantitative analyses [15]. As expected, we found that LPS treatment enhanced the production of NO by BV2 cells. The nitrite concentration measured in cell supernatants was 1.53 ± 0.08 µM for non-stimulated cells (control) and 4.52 ± 0.12 µM for LPS-activated BV2 cells (LPS, Table 1).

Cell viability was higher than 90% except for treatments with derivatives **4**, **5** and **6** that reduced cell viability ranged from 68% to 55%. That is in agreement with previous results in which is described that isatin derivatives substituted in position 5 showed antiproliferative activity on cancer cell lines [16,17]. In contrast, Matheus et al. reported that 5-chloro isatin did not reduce cell viability by more than 20% on mouse monocyte-macrophages at concentrations 10–100 µM [18]. Herein, we focused on treatments that ensured cell viability higher than 90% (Figure 1).

As shown in Figure 1, the reference compound isatin (**1**) did not show inhibition of NO release but rather a non-significant increase. While 5-fluor isatin (**2**) showed a slight inhibition of NO production by BV2 microglia cells (6%), chlorinated compounds **3** and **7** showed a significant reduction, being the 5-chloro isatine (compound **3**) the most effective in reducing NO release up to 49%. Compound **7**, substituted at the 4-position of the isatin with chloro, was less effective in reducing NO than its analogue substituted at the 5-position (compound **3**), despite having the same Log *p*-value. The exchange of Cl by Br at the 4-position of isatin did not improve NO inhibition. Finally, low inhibition (9%) was observed with the disubstitution of the aryl ring compound **9** (Table 1 and Figure 1).

These results point out that both the substitution at the 5-position of isatin and liphophilicity of the derivatives are key points for modulating their cytotoxic and anti-neuroinflammatory activity in BV2 microglia cells.

Based on the results, and aiming to obtain more potent anti-neuroinflammatory compounds, we synthesised a series of isatin derivatives and their 5-chlorinated analogous. We discussed the effect of different chains in position 1 as well as the substitution in position 3 on anti-neuroinflammatory activity, in comparison with the activity of the 5-chlorinated analogues.

### 2.2. Chemical Synthesis

A series of *N^1^*-alkylated isatin derivatives were easily prepared by conventional reactions following three methodologies depending on the functional group to be attached: (i) the direct alkylation of isatin **1** or 5-chloro isatin **3** with the corresponding haloalkane; (ii) the reaction of **1** or **3** with formaldehyde; and (iii) the reaction of **13** or **20** with succinic anhydride. On the other hand, C^3^ hydrazones derivatives were synthesised by condensation of isatin **1** or 5-chloro isatin **3** with the corresponding hydrazine (iv) as depicted in Figure 1. The C=N double bond in isatin hydrazones creates suitable conditions for the formation of two isomers, however, only the Z-isomer was isolated from the reaction mixture during synthesis through its stabilisation by intramolecular hydrogen bonding [19].

### 2.3. Biological Activity of Synthetic Isatin Derivatives

We evaluated the effect of the synthetic isatin derivatives on NO release by LPS-activated BV2 microglia cells and their effect on cell viability.

As shown in Table 2, most compounds were non-cytotoxic to BV2 microglia cells at 25 µM. However, compounds **17**, **18** and **19** with both substitutions at position 1 (*N^1^*-alkylated) and position 5 (chloro) of isatin reduced cell viability ranging from 85% to 67% (in accordance with the results observed for compounds **5** and **6**, Table 1). In contrast, their non-chlorinated analogues (compounds **10**, **11** and **12,** respectively) had no effect on the viability of BV2 microglia cells. We found that phenyl hydrazones (compounds **16** and **23**) also reduced cell viability by up to 83%. These results confirmed that cytotoxicity seems to be aligned with higher values of Log *p* (Table 2), although a change of the scaffold’s conformation cannot be ruled out, nor can the engagement of target proteins through halogen bonds due to the chlorine presence [20].

Figure 2 shows the results for derivatives with non-cytotoxic effects at 25 µM concentration, ensuring cell viabilities higher than 90%. We observed that all 5-chlorinated isatin derivatives were able to significantly reduce NO release in BV2 cells, except compound **21**, which showed a non-significant reduction (striped columns, Figure 2).

It is worth noting that *N^1^*-alkylated compounds **10**, **11**, **12**, **13** and **20** notably reduced NO release by LPS-stimulated BV2 microglia cells. Among them, **10** and **20** were the most efficient in reducing NO release, 2.6 and 3-fold, respectively (Table 2). Notably, chlorinated compound **20** showed the best results in reducing NO release up to 68% (*p* < 0.0001) compared to LPS-stimulated cells treated with a vehicle. However, no inhibition was observed with compounds **14** and **21** (Figure 2), indicating that *N*-substitutions with large chains and polar groups disfavour the anti-neuroinflammatory activity. These results point out that based on the substitution pattern, it is possible to enhance anti-neuroinflammatory activity. The presence of a chlorine atom at the aromatic ring or at the *N*-alkyl chain seems to be essential for the inhibition of NO release.

The effect of hydrazone formation at position 3 of isatin needs further studies as the phenyl hydrazones reduced cell viability while promising results were obtained with cyanoethylhydrazones (**15**, **22**), which were non-cytotoxic and capable to reduce NO release up to 19% and 44%, respectively (Figure 2).

As noted above, cell viability was reduced with derivatives **17**, **18**, **19** and phenylhydrazones **16** and **23** (Table 2). Although a low number of cells could be associated with lower NO concentration in cell media, it is worth noting that phenyhydrazone derivatives increased NO release, in particular, compound **23** showed a significant 2.5-fold NO increase (Table 2). It has been reported that inhibition of several cyclin-dependent kinases reduced LPS-induced NO production in macrophages, but not cyclin-dependent kinase 2 (CDK2) [21]. On the other hand, a series of isatin-hydrazones with cytotoxic activity in human adenocarcinoma cell lines showed CDK2 inhibitory activity [22]. Together, the results suggest that cyclin-dependent kinases may play important roles in the mechanisms by which synthetic compounds attenuate inflammation [22] and further studies are needed to understand the mechanism by which phenylhydrazones derivatives increase NO production in BV2 microglia cells.

We further examined the capability of isatin derivatives to reduce the production of proinflammatory mediators IL-6 and TNF-α, since their levels are increased in several neurodegenerative disorders associated with neuroinflammation [23]. The activation of microglial cells by LPS is associated with the NF-κB signalling pathway. The activation of transcription factor NF-κB produces proinflammatory cytokines and mediators such as IL-6 and TNF-α that affect neuronal receptors with an overactivation of protein kinases. These patterns of pathological events can be applied to several neurodegenerative disorders.

We measured cytokine IL-6 and TNF-α concentrations in BV2 cell cultures by microfluidic ELISA equipment ELLA. The effect of isatin derivatives on the production of IL-6 and TNF-α is shown in Figure 3.

Unstimulated cells released IL-6 at a concentration of 260 ± 28 pg/mL to cell media (Appendix A), while LPS activation led to a 10-fold increase in IL-6 concentration. We detected 2772 ± 521 pg/mL of IL-6 in LPS-activated BV2 cell media (LPS, Figure 3A), which is in agreement with recent results reported for LPS-activated BV2 cells [8]. Derivatives **10** and **20** were able to significantly reduce IL-6 concentration by 58% and 60% respectively (*p* < 0.05, Figure 3A). Cytokine IL-6 is a potent mediator of cellular communication and is crucial in the regulation of innate and adaptive inflammatory responses. Microglia respond strongly to IL-6, and production of IL-6 is upregulated in many chronic neuro-inflammatory diseases [24].

The results showed that LPS-activation of BV2 cells leads to 728 ± 60 pg/mL TNF-α concentration in cell media versus 206 ± 22 pg/mL for unstimulated BV2 cells (Figure 3B and Appendix A). The 3.5-fold increase in TNF-α production by LPS-activated BV2 cells is in agreement with the published work [10]. Pretreatment with compound **3** showed the most significant reduction in TNF-α concentration in LPS-activated BV2 cell media (62% decrease, *p* < 0.05, Figure 3B). As expected, significant reductions in TNF-α release by activated BV2 cells were also observed by pretreatments with derivatives **10** and **20**, which produced a 46% reduction in TNF-α concentration in cell media (*p* < 0.05, Figure 3B). There is growing evidence that elevated levels of IL6 and TNF-α in the brain are associated with neuroinflammation and neuronal damage, suggesting that targeting both IL-6 and TNF-α prevents not only central nervous system inflammation but also neuronal death [10].

## 3. Conclusions

In the present study, the effect of synthetic isatin derivatives on the reduction of pro-inflammatory mediators released by BV2 microglia cells has been evaluated for the first time. Through the optimisation of the substitutional groups in the isatin moiety, compounds **10** and **20** showed the best results at 25 µM on reduction of nitric oxide, pro-inflammatory interleukin 6 and tumour necrosis factor α release by BV2 microglia cells. Since it is required to develop new molecules with therapeutic potential in neurodegenerative disorders, we identified compounds **10** and **20** as lead candidates for further development of new anti-neuroinflammatory and neuroprotective agents and are worthy of further study.

## 4. Experimental Section

### 4.1. Materials and Methods

All chemicals used in this study were of reagent grade or higher and were purchased from commercial suppliers and used as received without further purification unless otherwise stated in the method. Solvents, reagents and isatins (**1–9**) were purchased from Scharlau (Barcelona, Spain), Acros Organics (Geel, Belgium), TCI (Portland, OR, US), Fluorochem (Hadfield, UK), Apollo Scientific (Manchester, UK), TRC (North York, ON, Canada), PanReac (Barcelona, Spain), AppliChem (Barcelona, Spain), Merck Millipore (Darmstadt, Germany), Alfa Aesar (Ward Hill, MA, US) and Aldrich (Darmstadt, Germany).

3-(4,5-dimethylthiazol-2-yl)-2,5-diphenyltetrazolium (MTT, Merck Millipore, (Darmstadt, Germany), Dulbecco’s Phosphate Buffered Saline (DPBS), Dulbecco’s High Glucose Modified Eagle’s Medium (DMEM) with and without Phenol Red, Fetal Bovine Serum (FBS) (Gibco™, Thermo Fisher Scientific, Madrid, Spain); L-glutamine saline solution (200 mM), penicillin-(10.000 U/mL)/streptomycin (10.000 µg/mL)/amphotericin B (25 µg/mL) solution (100×, Merck Millipore, (Darmstadt, Germany)); LPS (isotype 0111:B4 from Sigma-Aldrich, Darmstadt, Germany); 75 cm^2^ cell culture flasks (Grynia-Labbox, Barcelona, Spain); 96-well plates (Sarstedt, Nümbrecht, Germany); Syringe filters, 0.2 and 0.45 µm (FisherBrand, Shanghai, China); Injekt^®^ syringes of 2, 5 and 10 mL (BBraun, Melsungen, Germany).

Thin-layer chromatography (TLC) was performed on aluminium sheets 60 F254 Merck silica gel and compounds were visualised by irradiation with UV-light and/or by treatment with a solution of Ce_2_MoO_4_, KMnO_4_ or ninhydrine in *n*-butanol (3% of acetic acid), followed by heating. All synthetic products were purified by flash silica gel column chromatography using hand-packed columns, or Isolera^TM^ One Flash Chromatography Instrument (Biotage^®^, Uppsala, Sweden), using 10g Sfär Duo silica columns at a flow rate of 40 mL/min. Solvents were removed by rotavapor^®^ R-100 (Buchi, Flawil, Switzerland) or a Telstar LyoQuest 55 lyophilizer (Telstar^®^, Madrid, Spain). Nuclear Magnetic Resonance (NMR) spectra were recorded at room temperature on a Bruker DPX (^1^H, 300 MHz and ^13^C, 75 MHz) spectrometer using CDCl_3_, CD_3_OD, D_2_O or DMSO-*d_6_* as deuterated solvent. Chemical shifts (δ) are expressed in parts per million relative to internal tetramethylsilane. Coupling constant values (*J*) are reported in hertz (Hz), and spin multiplicities are indicated by the following symbols: s (singlet), d (doublet), t (triplet), q (quartet) and m (multiplet). The spectra were processed using MestReNova version 14.0 software (Mestrelab Research, Santiago de Compostela, Spain). Mass spectra (MS) were recorded on MALDI-TOF/TOF Bruker Ultraflex model or ESI-ion trap Bruker HCT Ultra mass spectrometer. High-performance liquid chromatography (HPLC) analyses were performed using an Agilent 1260 Infinity quaternary pump chromatograph, equipped with ultraviolet-visible Agilent detector and automatic injector system. Data acquisition and processing were accomplished with Agilent LC OpenLab CDS software (version 0.1.06). A reverse phase Kromaphase C18 (5 µm, 4.6 mm × 150 mm) column was used as the stationary phase, using a C18 guard cartridge. Isatin derivatives separation was carried out with a mobile phase of water-acetonitrile mixture with formic acid (0.1%) on an isocratic method 50–50 at 20 °C. The flow rate was 0.5 mL/min and the injection volume was 20 µL. The elution profile was monitored by recording the UV absorbance at 254 nm. The retention time of each compound (Rt) is given in minutes (min). All synthesised compounds were proven by this method to ≥95% purity. Stock solutions of isatin derivatives for cell assays were prepared in DMSO and diluted with DMEM at working concentrations.

### 4.2. Synthesis

#### 4.2.1. Synthesis of the *N*-Derivatives

*N*-derivatives (compounds **10**, **11**, **12**, **17**, **18** and **19**) were carried out following the protocol described by Chen et al. with mild modifications [25]: To a solution of isatin 1 or 5-chloro isatin 3 (3 mmol) in DMF (20 mL) under an argon atmosphere, K_2_CO_3_ (3 eq.) and the corresponding haloalkane (1.2 eq.) were added. The reaction mixture was stirred at room temperature for 12 h until the complete disappearance of 1 or 3 as evidenced by TLC. The solvent was removed under vacuum, and a solution of 1M NH_4_Cl_aq_. (10 mL) was added to the residue and extracted with ethyl acetate (10 mL × 3). The organic fractions were combined, dried over Na_2_SO_4_, filtered and concentrated. The crude product was purified by column chromatography (silica gel, hexane/ethyl acetate = 1:1 to 100% EtOAc).

#### 4.2.2. Synthesis of 1-(Hydroxymethyl)-Isatines

1-(hydroxymethyl)-isatines (compounds **13** and **20**) was performed according to the protocol described by Aboul-Fadl and coworkers [26]: A mixture of **1** or **3** (1,0 mmol) and formaldehyde (37 wt. % in H_2_O, 3 mL) in water (H_2_Omiliq, 10 mL) was stirred at reflux (100 °C) for 2 h until the disappearance of **1** or **3** was evidenced by TLC chromatography. The resulting solution was filtered while hot and the filtrate was kept at 4 °C overnight. The crystalline product formed was filtered, dried, and recrystallised from ethyl acetate.

#### 4.2.3. Synthesis of Isatin-Hydrazones

Isatin-hydrazones (**15**, **16**, **22** and **23**) were carried out following the protocol described by Bekircan and Bektas, modifying the reactants [27]: A mixture of **1** or **3** (3.73 mmol) with the corresponding hydrazine (1.2 eq.) and acetic acid (200 µL) in ethanol (20 mL) was refluxed (80 °C) for 3.5 h until disappearance of **1** or **3** was observed by TLC. The solvent was removed in vacuo and the residue was purified by silica gel column chromatography (hexane/acetone = 3:1 to 0:1).

#### 4.2.4. Synthesis of Target Compounds

##### 1-(3-chloropropyl)-1H-indole-2,3-dione (**10**)

Compound **10** was prepared as described in 4.2.1 using **1** (441 mg, 3 mmol) and 1-bromo-3-chloropropane (360 µL, 3.6 mmol) to obtain the desired product (600 mg, 89% yield) as an orange crystal. HPLC Rt = 10.60 min. ^1^H-NMR was in agreement with those reported in the literature [28]. ^1^H-NMR (300 MHz, CDCl_3_) *δ* 7.64–7.59 (m, 2H), 7.13–7.08 (m, 1H), 6.89 (d, *J* = 8.5 Hz, 1H), 3.91 (t, *J* = 7.0 Hz, 2H), 3.63 (t, *J* = 6.1 Hz, 2H), 2.25–2.16 (m, 2H). MS (ESI+): *m*/*z* calcd for C_11_H_10_ClNNaO_2_: 246.0; found: 245.6 [M + Na]^+^.

##### 1-(pent-4-en-1-yl)-1H-indole-2,3-dione (**11**)

Compound **11** was prepared as described in 4.2.1 using isatin **1** (147 mg, 1 mmol) and 5-bromo-1-pentene (141.9 µL, 1.2 mmol) to obtain the desired product (215 mg, 95% yield) as a dark orange crystal. HPLC Rt: 15.55 min. ^1^H-NMR was in agreement with those reported in the literature [29]. ^1^H-NMR (300 MHz, CDCl_3_) *δ* 7.60–7.55 (m, 2H), 7.10 (t, *J* = 7.5 Hz, 1H), 6.88 (d, *J* = 7.8 Hz, 1H), 5.87–5.74 (m, 1H), 5.10–4.99 (m, 2H), 3.72 (t, *J* = 7.4 Hz, 2H), 2.19–2.12 (m, 2H), 1.85–1.75 (m, 2H). MS (ESI+); *m*/*z* calcd for C_13_H_13_NNaO_2_: 238.0; found: 237.6 [M + Na]^+^.

##### 1-butyl-1H-indole-2,3-dione (**12**)

Compound **12** was prepared as described in 4.2.1 using isatin (1 g, 6.8 mmol) and 1-bromobutane (876 µL, 8.16 mmol) to obtain the desired product (1.33 g, 96%) as an orange crystal. HPLC Rt: 14.26 min. ^1^H-NMR was in agreement with those reported in the literature [30]. ^1^H-NMR (300 MHz, CDCl_3_) *δ* 7.61–7.55 (m, 2H), 7.13–7.08 (m, 1H), 6.91–6.88 (m, 1H), 3.72 (t, *J* = 7.3 Hz, 2H), 1.71–1.63 (m, 2H), 1.45–1.37 (m, 2H), 0.97 (t, *J* = 7.3 Hz, 3H). MS (ESI+): *m*/*z* calcd for C_12_H_13_NNaO_2_: 226.0; found: 225.6 [M + Na]^+^.

##### 1-(hydroxymethyl)-1H-indole-2,3-dione (**13**)

Compound **13** was prepared as described in 4.2.2 using **1** (1.47 g, 1,0 mmol) to obtain the product (1.47 g, 83% yield) as an orange crystal. HPLC Rt: 4.55 min. ^1^H-NMR was in agreement with those reported in the literature [26]. ^1^H-NMR (300 MHz, DMSO-d_6_) δ 7.70 (ddd, J = 8.0, 7.5, 1.4 Hz, 1H), 7.58 (ddd, J = 7.5, 1.4, 0.6 Hz, 1H), 7.25 (dt, J = 8.0, 0.7 Hz, 1H), 7.17 (td, J = 7.5, 0.9 Hz, 1H), 6.41 (t, J = 7.1 Hz, 1H), 5.09 (d, J = 7.1 Hz, 2H). ^13^C NMR (75 MHz, DMSO-d_6_) δ 183.5 (C), 157.6 (C), 150.3 (C), 138.3 (CH), 124.5 (CH), 123.5 (CH), 117.3 (C), 111.7 (CH), 62.9 (CH_2_). MS (ESI+): *m*/*z* calcd. for C_9_H_7_NNaO_3_: 200.0; found: 199.5 [M + Na]^+^.

##### 4-[(2,3-dioxo-2,3-dihydro-1H-indole-1-yl)methoxy]-4-oxobutanoic Acid (**14**)

**13** (218 mg, 1.23 mmol) was dissolved in 5 mL of pyridine. The mixture was cooled at 0 °C and succinic anhydride (307 mg, 3.07 mmol) was added. The reaction mixture was stirred at room temperature for 24 h. The reaction was quenched by the addition of 5 mL of water and extracted with ethyl acetate (3 × 10 mL), the organic phase was washed with a saturated solution of NaHCO_3aq_. After drying with MgSO_4_, the solvent was removed by vacuum and the residue was purified by column chromatography to remove pyridine residues (silica gel, hexane/acetone = 1:1 to 0:1) to obtain **14** (197 mg, 57% yield) as an orange solid. HPLC Rt: 4.77 min. ^1^H-NMR (300 MHz, DMSO-d_6_) δ 7.72–7.68 (m, 1H), 7.63–7.60 (m, 1H), 7.28 (d, J = 8.0 Hz, 1H), 7.21 (t, J = 7.6 Hz, 1H), 5.72 (s, 2H), 2.57–2.41 (m, 4H, overlapping with solvent signal DMSO). ^13^C-NMR (75 MHz, DMSO-d_6_) δ 182.0 (C), 173.3 (C), 171.7(C), 158.0 (C), 149.2 (C), 138.3 (CH), 124.6 (CH), 124,0 (CH), 117.7 (C), 111.5 (CH), 63.3 (CH_2_), 28.9 (CH_2_), 28.6 (CH_2_). MS (ESI+): m/z calcd. for C_13_H_11_NNaO_6_: 300.2; found: 299.7 [M + Na]^+^.

##### 3-[(2Z)-2-(2-oxo-1,2-dihydro-3H-indole-3-ylidene)hydrazinyl]propanenitrile (**15**)

Compound **15** was prepared as described in 4.2.3 using **1** (500 mg, 3.39 mmol) and 2-cyanoethylhydrazine (305 µL, 3.73 mmol) to obtain the desired product (623 mg, 85% yield) as a yellow crystal. HPLC Rt: 6.46 min. ^1^H-NMR (300 MHz, DMSO-d_6_) δ 11.00 (t, J = 4.5 Hz, 1H), 10.77 (s, 1H), 7.36 (dd, J = 7.5, 1.2 Hz, 1H), 7.16 (td, J = 7.7, 1.2 Hz, 1H), 6.98 (td, J = 7.5, 0.8 Hz, 1H), 6.87 (dt, J = 7.8, 0.8 Hz, 1H), 3.81–3.75 (m, 2H), 2.88 (t, J = 6.4 Hz, 2H). ^13^C-NMR (75 MHz, DMSO-d_6_) δ 162.8 (C), 138.8 (C), 127.31 (CH), 125.7 (C), 121.8 (C), 121.5 (CH), 119.1 (C), 117.7 (CH), 110.1 (CH), 46.8 (CH_2_), 18.1 (CH_2_). MS (MALDI-TOF +): calcd. for C_11_H_11_N_4_O: 215.09; found: 215,11 [M + H]^+^.

##### (3Z)-3-(2-phenylhydrazinylidene)-1,3-dihydro-2H-indole-2-one (**16**)

Compound **16** was prepared as described in 4.2.3 using **1** (205 mg, 1.3 mmol) and phenylhydrazine (151 µL, 1.4 mmol) to obtain the desired product (244 mg, 74% yield) as a yellow powder. HPLC Rt: 7.36 min. ^1^H-NMR was in agreement with those reported in the literature [19]. ^1^H-NMR (300 MHz, DMSO-d_6_) δ 12.75 (s, 1H), 11.02 (s, 1H), 7.57–7.54 (m, 1H), 7.45–7.34 (m, 4H), 7.25 (td, J = 7.7, 1.3 Hz, 1H), 7.08–7.02 (m, 2H), 6.92 (dt, J = 7.8, 0.9 Hz, 1H). MS (ESI+): *m*/*z* calcd. for C_14_H_11_N_3_NaO: 260.0; found: 259.7 [M + Na]^+^.

##### 5-chloro-1-(3-chloropropyl)-1H-indole-2,3-dione (**17**)

Compound **17** was prepared as described in 4.2.1, using **3** (200 mg, 1.1 mmol) and 1-bromo-3-chloropropane (130 µL, 1.32 mmol) to obtain the desired product (208 mg, 73% yield) as dark orange crystal. HPLC Rt = 18.25 min. ^1^H-NMR (300 MHz, CDCl_3_) *δ* 7.58–7.55 (m, 2H), 7.00–6.96 (m, 1H), 3.89 (t, *J* = 7.0 Hz, 2H), 3.61 (t, *J* = 6.0 Hz, 2H), 2.22–2.13 (m, 2H). ^13^C-NMR (75 MHz, CDCl_3_) *δ* 182.3 (C), 157.9 (C), 149.1 (C), 138.0 (CH), 129.8 (C), 125.6 (CH), 118.5 (C), 111.5 (CH), 42.0 (CH_2_), 38.0 (CH_2_), 30.2 (CH_2_). MS (ESI+): *m*/*z* calcd for C_11_H_9_Cl_2_NNaO_2_: 279.9; found: 279.6 [M + Na]^+^.

##### 5-chloro-1-(pent-4-en-1-yl)-1H-indole-2,3-dione (**18**)

Compound **18** was prepared as described in 4.2.1, using **3** (300 mg, 1.65 mmol) and 5-bromo-1-pentene (234 µL, 1.98 mmol) to obtain the desired product (344 mg, 84% yield) as a dark orange crystal. HPLC Rt: 28.02 min. ^1^H-NMR (300 MHz, CDCl_3_) *δ* 7.55–7.52 (m, 2H), 6.87–6.84 (m, 1H), 5.85–5.72 (m, 1H), 5.09–4.99 (m, 2H), 3.71 (t, *J* = 7.4 Hz, 2H), 2.18–2.09 (m, 2H), 1.83–1,73 (m, 2H). ^13^C-NMR (75 MHz, CDCl_3_) *δ* 182.6 (C), 157.7 (C), 149.3 (C), 137.8 (CH), 136.8 (CH), 129.6 (C), 125.4 (CH), 118.5 (C), 116.1 (CH_2_), 111.5 (CH), 39.9 (CH_2_), 30.9 (CH_2_), 26.2 (CH_2_). MS (ESI+): *m*/*z* calcd for C_13_H_12_ClNNaO_2_: 272.1; found: 271.6 [M + Na]^+^.

##### 1-butyl-5-chloro-1H-indole-2,3-dione (**19**)

Compound **19** was prepared as described in 4.2.1, using **3** (200 mg, 1.1 mmol) and 1-bromobutane (142 µL, 1.32 mmol) to obtain the desired product (251 mg, 96% yield) as an orange crystal. HPLC Rt: 25.90 min. ^1^H-NMR was in agreement with those reported in the literature [31]. ^1^H-NMR (300 MHz, CDCl_3_) *δ* 7.56–7.52 (m, 2H), 6.88–6.83 (m, 1H), 3.71 (t, *J* = 7.3 Hz, 2H), 1.72–1.62 (m, 2H), 1.46–1.34 (m, 2H), 0.96 (t, *J* = 7.3 Hz, 3H). MS (ESI+): *m*/*z* calcd for C_12_H_12_ClNNaO_2_: 260.0; found: 259.6 [M + Na]^+^.

##### 5-cloro-1-(hydroxymethyl)-1H-indole-2,3-dione (**20**)

Compound **20** was prepared as described in 4.2.2 using **3** (190 mg, 1.0 mmol) to obtain the desired product (190 mg, 86% yield) as an orange crystal. HPLC Rt: 5.64 min. ^1^H-NMR (300 MHz, DMSO-d_6_) δ 7.75 (dd, J = 8.5, 2.3 Hz, 1H), 7.63 (dd, J = 2.3, 0.5 Hz, 1H), 7.28 (dd, J = 8.4, 0.5 Hz, 1H), 6.45 (t, J = 7.1 Hz, 1H), 5.08 (d, J = 7.1 Hz, 2H). ^13^C-NMR (75 MHz, DMSO-d_6_) δ 182.4 (C), 157.4 (C), 148.8 (C), 137.2 (CH), 127.8 (C), 124.0 (CH), 118.7 (C), 113.5 (CH), 63.1 (CH_2_). MS (ESI+): *m*/*z* calcd. for C_9_H_6_ClNNaO_3_: 233.9; found: 233.5 [M + Na]^+^.

##### 4-[(5-chloro-2,3-dioxo-2,3-dihydro-1H-indole-1-yl)methoxy]-4-oxobutanoic Acid (**21**)

**20** (179 mg, 0.84 mmol) was dissolved in pyridine (3.4 mL). The mixture was cooled at 0 °C and succinic anhydride (307 mg, 3.07 mmol) was added. The mixture was stirred at room temperature for 24 h. The reaction was quenched by the addition of 5 mL of water and extracted with ethyl acetate (3 × 10 mL), the organic phase was washed with a saturated solution of NaHCO_3 aq_. After drying with MgSO_4_, the solvent was removed by vacuum and the residue was purified by column chromatography to remove pyridine residues (silica gel, hexane/acetone = 1:1 to 0:1) to obtain **21** (237 mg, 91% yield) as an orange solid. HPLC Rt: 6.45 min. ^1^H-NMR (300 MHz, DMSO-d_6_) δ 7.77 (dd, J = 8.5, 2.3 Hz, 1H), 7.68 (d, J = 2.1 Hz, 1H), 7.33 (d, J = 8.5 Hz, 1H), 5.72 (s, 2H), 2.58–2.45 (m, 4H, overlapping with solvent signal DMSO). ^13^C-NMR (75 MHz, DMSO-d_6_) δ 181.1 (C), 173.5 (C), 171.8 (C), 157.8 (C), 147.8 (C), 137.2 (CH), 128.4 (C), 124.1 (CH), 119.2 (C), 113.5 (CH), 63.2 (CH_2_), 28.7 (CH_2_), 28.6 (CH_2_). MS (ESI+): *m*/*z* calcd. for C_13_H_10_ClNNaO_6_: 334.0; found: 333.7 [M + Na]^+^; 644.7 [2M + Na]^+^.

##### 3-[(2Z)-2-(5-chloro-2-oxo-1,2-dihydro-3H-indole-3ylidene)hydrazinyl] propanenitrile (**22**)

Compound **22** was prepared as described in 4.2.3, using **3** (200 mg, 1.1 mmol) and 2-cyanoethylhydrazine (130 µL, 1.6 mmol) to obtain the desired product (138 mg, 51% yield) as a yellow powder. HPLC Rt: 9.79 min. ^1^H-NMR (300 MHz, DMSO-d_6_) δ 11.10 (t, J = 4.5 Hz, 1H), 10.89 (s, 1H), 7.31 (d, J = 2.2 Hz, 1H), 7.18 (dd, J = 8.3, 2.2 Hz, 1H), 6.87 (d, J = 8.3 Hz, 1H), 3.81–3.76 (m, 2H), 2.91 (t, J = 6.4 Hz, 2H). ^13^C-NMR (75 MHz, DMSO-d_6_) δ 162.6 (C), 137.3 (C), 126.6 (CH), 125.7 (C), 124.4 (C), 123.56 (C), 119.1 (C), 117.1 (CH), 111.5 (CH), 47.0 (CH_2_), 18.0 (CH_2_). MS (ESI+): *m*/*z* calcd. for C_11_H_9_ClN_4_NaO: 271.0; found: 270.6 [M + Na]^+^.

##### (3Z)-5-chloro-3-(2-phenylhydrazinylidene)-1,3-dihydro-2H-indole-2-one (**23**)

Compound **23** was prepared as described in 4.2.3, using **3** (200 mg, 1.1 mmol) and phenylhydrazine (108.5 µL, 1.1 mmol) to give the desired product (235 mg, 78% yield) as a yellow powder. HPLC Rt: 19.11 min. ^1^NMR was in agreement with those reported in the literature [32]. ^1^H-NMR (300 MHz, DMSO-d_6_) δ 12.75 (s, 1H), 11.13 (s, 1H), 7.57 (dd, J = 2.2, 0.5 Hz, 1H), 7.50–7.47 (m, 2H), 7.41–7.35 (m, 2H), 7.26 (dd, J = 8.3, 2.2 Hz, 1H), 7.10–7.05 (m, 1H), 6.93 (dd, J = 8.3, 0.5 Hz, 1H). MS (ESI+): *m*/*z* calcd. for C_14_H_10_ClN_3_NaO: 294.0; found: 293.7 [M + Na]^+^.

### 4.3. Cell Culture

Microglia BV2 cell line was obtained from Interlab Cell Line Collection (ICLC), number ATL 03,001 (Genova, Italy). BV2 microglia cells were cultured in high glucose DMEM media supplemented with 100U/mL penicillin, 100 μg/mL streptomicyn, 0.25 μg/mL amphotericin B and 2 mM L-glutamine, plus 10% fetal bovine serum (FBS) previously heat-inactivated (30 min, 56 °C), in 75-cm^2^ flasks at 37 °C under atmosphere of 5% CO_2_ and 95% air. After reaching confluency, cells were dissociated (three times per week) using 0.025% trypsin and 0.01% EDTA in DPBS.

### 4.4. Cell Viability Assay

Cell viability was assessed using the MTT assay for mitochondrial dehydrogenase activity, as previously described [33]. Briefly, BV2 cells were seeded in a 96-well plate at a density of 7.5 × 10^3^ cells/well and grown in supplemented DMEM with 10% FBS for 24 h. Then, media was removed, and cells were cultured in supplemented DMEM without phenol red nor FBS, with different concentrations of isatin derivatives (ranging from 1 μM to 500 μM) for 24 h. After incubation, supernatants were removed and cells were treated with MTT (0.5 mg/mL in DPBS:DMEM 1:10, 100 µL) for 3 h. Then, MTT excess was carefully removed and DMSO (100 µL) was added to each well to dissolve the formazan crystals. The absorbance was measured at 562 nm on a spectrophotometer plate reader (Sunrise^TM^, TECAN). The control groups consisted of cells treated with DMEM plus vehicle used to dissolve isatin derivatives and was considered 100% of viable cells. The percentage of viability and inhibition was calculated as follows: % Viability = (Absorbance_562_ of treated cells/Absorbance_562_ of control cells) × 100; % Inhibition = 100-% viability.

### 4.5. LPS-Activation of BV2 Cells

BV2 cells were seeded in a 96-well plate at a density of 15 × 10^3^ cells/well and grown in supplemented DMEM with 10% FBS for 24 h. Then, media was removed, and cells were cultured in supplemented DMEM without phenol red nor FBS, and pretreated with isatin derivatives (2×, 75 µL) for 1 h. Then, LPS (isotype 0111:B4) in DMEM without phenol red nor FBS was added (2×, 75 µL) and cells were incubated for 24 h. The final concentration of compounds was 5, 25 or 50 µM, and the concentration for LPS stimulation was 1 µg/mL. After incubation, an aliquot of supernatants (75 µL) was taken for nitric oxide determination, and another aliquot of supernatants (75 µL) was kept at −80 °C until IL-6 and TNF-α quantification. Cells were incubated with MTT for determination of their viability after treatments.

### 4.6. Nitric Oxide Release Determination

The release of nitric oxide (NO) to the cultured medium by BV2 was indirectly determined by measuring nitrite, which is formed by the oxidation of NO, using the modified Griess reagent (naphthylethylenediamine dihydrochloride and sulphanilamide in phosphoric acid) as previously described [34]. Briefly, the nitrite reacts with the acid solution of sulphanilamide to form a diazonium salt, which couples with naphthylethylenediamine to form a pink/purple diazo dye whose absorbance can be measured at 562 nm. To determine the nitrite concentration in cell culture media, a stock solution of Griess reagent in miliQ-H_2_O (40 mg/mL) was prepared and mixed with cell culture media (1:1) in a 96-plate (150 µL/well) and incubated in the dark for 15 min. Then, absorbance was measured at 562 nm in an ELISA plate reader (Sunrise™; TECAN). Sodium nitrite (NaNO_2_) was used to generate a standard curve with concentrations ranging from 50 to 0.78 µM, and nitrite concentration in cell-cultured media was determined by comparison of their absorbance values with those of the standard curve.

### 4.7. Measurement of IL-6 and TNF-α Levels

Supernatans collected and stored at −80 °C from BV2 cell cultures treated with compounds and then stimulated with LPS (1 µg/mL), were analysed using the microfluidic ELISA equipment ELLA-Protein Simple (Biotechne, Minneapolis, MN, USA). Concentrations of IL-6 cytokine and TNF-α released by BV2 cells after treatments were measured. Supernatant samples were diluted according to the manufacturer’s instructions for IL-6 and TNF-α determination. The upper and lower limits of quantification (ULOQ and LLOQ, respectively) were calculated according to the manufacturer’s instructions. The limits of quantification for the IL-6 and TFN-α were calculated from the limit of quantification of the ELLA equipment multiplied by 2 and 100 respectively, taking into account the dilution of the sample. For IL-6 the ULOQ was 11,540 pg/mL, and the LLOQ was 1.2 pg/mL. For TFN-α, the ULOQ was 29,300 pg/mL, and the LLOQ was 31 pg/mL. The results are the mean of three independent experiments performed in triplicate.

### 4.8. Statistical Analysis

Results were expressed as mean ± standard error of the mean (SEM) of at least three independent experiments performed in triplicate. GraphPad Prism software, version 9.0 (GraphPad Software, Inc.) was used for statistical analysis and to create the graphs. Statistically significant differences [35,36] between multiple groups were determined by Kruskal–Wallis test. To compare IL-6 and TNF-α concentrations with the LPS group, two group comparisons Mann–Whitney test was used. Differences between the datasets were accepted as significant when *p* value < 0.05 and significant differences are depicted as follows: * *p* < 0.05, ** *p* < 0.01, *** *p* < 0.001, **** *p* < 0.0001.

## Data Availability

Not applicable.

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
