# Peer review of "Evaluation of Anti-Neuroinflammatory Activity of Isatin Derivatives in Activated Microglia"

_molecules, 2023, doi:10.3390/molecules28124882_

Round 1
Reviewer 1 Report
This manuscript describes novel data obtained by using newly synthesized compounds.
I am not qualified to assess the organic synthesis protocols and quality control of the novel derivatives; therefore, I am not commenting on this aspect of the study..
The description of the biological testing of compounds could be considered for publication after the following issues have been addressed
Major issues:
1. None of the tables show statistically significant values; therefore, the apparent trends seen in these tables cannot be referred to as “effects”.
2. Cell viability values are only shown in tables; therefore, there are no statistically validated data proving that some of the compounds are not toxic at 25 uM. Provide viability data similar to those shown in figures.
3. Figure 3 and table 3 appear to show the same data. Eliminated the duplication.
4. Why only 25 uM concentration was tested? The interpretation of the biological activity of the compounds could be quite different if higher or lower concentrations are tested. Concentration-dependent effects in a broader range of compounds must be studied and presented for all identified “active and non-toxic” compounds, and for at least several “non-active and non-toxic” compounds. To illustrate: the abstract sates that compounds 10 and 20 showed the best results – the fact that only one concentration was used must be disclosed. In addition, re-testing all compounds at a different concentration will likely lead to a different conclusion.
5. The manuscript states that isatin derivatives are known to affect cell proliferation. Was the reduction of MTT signal observed in this study in the presence of derivatives due to reduced proliferation rate of cells, due to direct toxicity (killing) of BV-2 cells, or both? These two very broad possibilities should be assessed experimentally.
6. BV-2 cells may be good models of murine neuroinflammatory response, but they likely are not adequate models of human neuroinflammatory response. For example, human microglia (and most monocytic cells) cannot be induced by LPS (an most other stimuli) to produce NO. This aspect of the model cells used in this study should be disclosed and discussed in the manuscript
Minor comments
Figure legends should only explain those *, **, etc. that are actually present on figures
minor editing is needed
Author Response
Dear Reviewer,
Thank you very much for your comments about the manuscript with ID: molecules-2436202 about Evaluation of Anti-Neuroinflammatory Activity of Synthetic Isatin Derivatives in Activated Microglia.
Please, find enclosed our revised manuscript and supplementary materials version according to your comments. Major revisions were performed in the existing manuscript, which are marked in red with the MS Word “Track Changes” function.
We have addressed the following issues in the revised manuscript:
- Statistically significant differences are showed in all the Tables and Figures of the revised manuscript, including statistically validated data for cell viability.
- Table and Figure legends have been revised and only the significant differences that are present on the corresponding Table or Figure are explained.
- Table 3 has been removed from the manuscript and included at the supporting information as Table S2.
- Concentrations of 5, 25 and 50 µM of all compounds were tested for cell viability and inhibition of LPS-induced NO release. These results have been included at the supplementary materials in Table S1. In the manuscript we focused the discussion of compounds activity at 25 µM because this concentration showed more relevant NO reductions than at 5 µM for treatments that ensured cell viability higher than 90% (page 3), and at 50 µM we observed lower cell viability for several compounds.
- Compounds 10 and 20 also showed the best results at 5 µM in reducing NO and 20 was the best at 50 µM with cell viability higher than 90%. However, as we focused the discussion at 25 µM, we disclosed this concentration in the abstract and conclusions. We corrected in page 1: “microglial activation” and in page 3: “cell viability”.
- An overview of in vitro methods to study microglia (https://doi.org/10.3389/fncel.2018.00242) describes that exposure of primary human microglia to LPS induced the secretion of IL-6 and TNF-α, however we agree there is no evidence of LPS-induction of NO, and we disclosed in page 2 that we use LPS-activated BV-2 as celular model of murine neuroinflammation.
Reviewer 2 Report
The manuscript by Cenalmor et al. investigated the anti-neuroinflammatory roles of Isatin derivatives in LPS-induced activated Microglia and authors have shown promising results on reduction of release of nitric oxide, pro-inflammatory IL6 and TNF α.
The introduction, material and methods were well written. However, reviewer was particularly concerned about the statistical data analysis. Although the statistical analysis was conducted using parametric and non-parametric approaches, given such a small sample size, reviewer recommends that the results should be analyzed by a non-parametric approach. Please refer to ARRIVE guidelines, and also other studies which could be referenced here, for example doi: 10.1016/j.neuropharm.2018.08.037.
In the discussion part, I suggest to better discuss the involvement Isatin on the mechanisms of both anti- and pro-inflammatory processes of microglia in diseases (https://www.ncbi.nlm.nih.gov/pmc/articles/PMC8950263/)
Author Response
Dear Reviewer,
Thank you very much for your comments about the manuscript with ID: molecules-2436202 about Evaluation of Anti-Neuroinflammatory Activity of Synthetic Isatin Derivatives in Activated Microglia.
Please, find enclosed our revised manuscript and supplementary materials version according to your comments. Major revisions were performed in the existing manuscript, which are marked in red with the MS Word “Track Changes” function.
We have addressed the following issues in the revised manuscript:
- Statistical analysis were conducted using non-parametric approaches and it has been accordingly described on page 14, at the Experimental Section, 4.8 Statistical Analysis.
- The statistics on all Tables and Figures have been revised and corrected accordingly.
- Thanks for the suggestion about the ARRIVE guidelines (Animal Research: Reporting of In Vivo Experiments), we will follow the recommendations to improve the reporting of research involving in vivo experiments in the future.
- The therapeutic outcomes of isatin derivatives against multiple diseases has been reviewed in reference 8 (page 2). However, their anti-inflammatory potential only has been described in few works aimed to evaluate general inflammation, not specifically neuroinflammation. This comment has been included in the discussion (page 2).
Reviewer 3 Report
Τhe presented research is interesting and well written.
However some more in terms of SAR should be inserted in the discussion explaining better the role of the physicochemical properties.
The scientific question is the evaluation of new neuroprotective agents. Isatin derivatives presented low cytotoxicity and reducing activity in the the release of nitric oxide, pro-inflammatory interleukin 6 and tumour necrosis factor α by microglial cells, leading to neuroprotection. This finding in new for this chemical group and innovative.
The compounds are acting on overactivated microglia cause neurotoxicity and prolong the inflammatory response in many neuropathologies.
The results are well presented and given within the Figures and the Tables with their statistical evaluation. The chemical synthesis and in vitro assays are well performed. The given conclusions are consistent with the results and explain the the main scientific question.
Some more reference for the biological history of isatins should be inserted.
Author Response
Dear Reviewer,
Thank you very much for your comments about the manuscript with ID: molecules-2436202 about Evaluation of Anti-Neuroinflammatory Activity of Synthetic Isatin Derivatives in Activated Microglia.
Please, find enclosed our revised manuscript and supplementary materials version according to your comments. Major revisions were performed in the existing manuscript, which are marked in red with the MS Word “Track Changes” function.
We have addressed the following issues in the revised manuscript:
- More discussion about SAR has been added in page 6.
- References 4 and 5 about biological history of isatins have been added at page 1.